# Gold Nanoparticles as a Platform for Delivery of Immunogenic Peptides to THP-1 Derived Macrophages: Insights into Nanotoxicity

**DOI:** 10.3390/vaccines13020119

**Published:** 2025-01-24

**Authors:** Eduardo Zúñiga, Braulio Contreras-Trigo, Jorge Buchert, Fabián Sáez-Ahumada, Leonardo Hernández, Víctor Fica-León, Estefania Nova-Lamperti, Bostjan Kobe, Fanny Guzmán, Víctor Diaz-García, Enrique Guzmán-Gutiérrez, Patricio Oyarzún

**Affiliations:** 1Facultad de Ingeniería, Arquitectura y Diseño, Universidad San Sebastián, Concepción 4081339, Chile; ezunigaz@docente.uss.cl (E.Z.); bcontrerast@docente.uss.cl (B.C.-T.); vfical@correo.uss.cl (V.F.-L.); victor.diazg@uss.cl (V.D.-G.); 2Departamento de Bioquímica Clínica e Inmunología, Facultad de Farmacia, Universidad de Concepción, Concepción 4070386, Chile; jbuchert2016@udec.cl (J.B.); fabsaez@udec.cl (F.S.-A.); lehernandez2019@udec.cl (L.H.); enova@udec.cl (E.N.-L.); eguzman@udec.cl (E.G.-G.); 3School of Chemistry and Molecular Biosciences, Institute for Molecular Bioscience, and Australian Infectious Diseases Research Centre, The University of Queensland, Brisbane, QLD 4072, Australia; b.kobe@uq.edu.au; 4Núcleo Biotecnología Curauma, Pontificia Universidad Católica de Valparaíso, Valparaíso 2373223, Chile; fanny.guzman@pucv.cl

**Keywords:** peptide-based nanovaccination, epitope-based vaccination, gold nanoparticle, T-cell epitope, nanotoxicity, cathepsin B activity

## Abstract

Background: Peptide-based nanovaccines have emerged as a promising strategy for combating infectious diseases, as they overcome the low immunogenicity that is inherent to short epitope-containing synthetic peptides. Gold nanoparticles (AuNPs) present several advantages as peptide nanocarriers, but a deeper understanding of the design criteria is paramount to accelerate the development of peptide-AuNPs nanoconjugates (p-AuNPs). Methods: Herein, we synthesized and characterized p-AuNPs of 23 nm (p-Au23) and 68 nm (p-Au68) with varying levels of peptide surface coverage and different peptide designs, investigating their effect on the cell viability (cell death and mitochondrial activity), cellular uptake, and cathepsin B activity in THP-1 macrophages. Results: p-Au23 proved no negative effect in the cell viability and high levels of nanoconjugate uptake, but p-Au68 induced strong toxicity to the cell line. The peptide sequences were successfully designed with spacer regions and a cell-penetrating peptide (pTAT) that enhanced cellular uptake and cathepsin B activity for p-Au23, while pTAT induced severe effects in the THP-1 viability (~40–60% cell death). Conclusions: These findings provide valuable insight into the design criteria of AuNPs and immunogenic peptides, along with nanotoxicity effects associated with AuNP size and surface charge in human monocyte-derived macrophages.

## 1. Introduction

Conventional vaccination strategies based on live-attenuated or whole-inactivated viral vaccines are slow to develop and have biosafety issues that make them poorly suited to respond to a rapidly evolving pandemic crisis [1]. Therefore, there is a strong demand for next-generation vaccination approaches enabling more effective, safer, and faster vaccines to better respond to emerging viral pathogens [2]. A promising approach is to focus vaccine development on the antigenic determinant regions specifically recognized by T-cells (T-cell epitopes). The so-called “epitope-based approach” focuses on short peptide fragments to elicit long-lasting protective immune responses, with the advantage of excluding potentially deleterious biocomponents from viruses (allergenic and/or immunosuppressive sequences) [3]. T-cell epitopes can be presented to T-cells in the context of major histocompatibility complex (MHC) proteins, which lead to stimulation of cellular immune responses (a phenomenon called MHC restriction). MHC class I and class II proteins present peptide antigens to CD8+ cytotoxic T-cells (CD8+ T-cell epitopes) and to CD4+ T-helper cells (CD4+ T-cell epitopes), respectively.

Epitope-based peptide vaccines offer the prospect of a more prominent role of HLA-restricted immune responses inducing large repertoires of T-cell specificities and inducing multi-strain protection against rapidly mutating viral pathogens. In addition, the rational selection of CD4+ and CD8+ T-cell epitopes in vaccine formulations allows the targeting of viral vulnerabilities and achieving precise control over immune response activation by stimulating different subpopulations of lymphocytes [3]. A large body of evidence points to CD8+ cytotoxic T-cells as key players in the immune response to intracellular infections [4]. However, the inclusion of CD4+ T-cell epitopes is considered essential for providing cognate help and inducing vigorous immune responses. These epitopes promote the expansion of cytotoxic T-cells as well as the generation and maintenance of T-cell memory and production of neutralizing antibodies by promoting B-cell differentiation into plasma cells [5,6]. In this context, immunoinformatics tools have become indispensable in aiding the high-throughput (genome-level) discovery of T-cell epitopes with broad or population-driven MHC coverage [7,8].

Peptides are, in general, poorly immunogenic and need to be delivered with additional immune-stimulating agents, such as adjuvants or particulate delivery systems/carriers that can boost their recognition by the immune system [9]. In this regard, nanotechnology-based approaches allow the increase in size of vaccine components to a virus-like scale, offering the opportunity for tunable size, optimal shape, and surface-anchored targeting epitopes that can dramatically enhance the immunogenicity of molecular vaccines [10,11]. Peptide conjugation to nanoparticles (NPs) also allows for surface display of multiple copies of the same peptide antigen (multivalency), mimicking spatially repetitive structures in pathogens (antigen recurrence) in a way that improves their engagement with the immune system [12]. In addition, the particulate form of the antigen enhances antigen-presenting cell (APC) sensing and cellular uptake with respect to the soluble form while also increasing persistence at the administration site and even acting as an immune adjuvant [13,14]. Consequently, there is a growing interest in the scientific community to explore the use of NP-based peptide vaccines for innovative designs and vaccination strategies against emerging pathogens [15,16].

Gold nanoparticles (AuNPs) have been proposed as ideal vehicles for drug delivery due to their straightforward synthesis [17], biocompatibility, non-toxic nature, controllable size and shape, as well as ease of surface biofunctionalization [18]. Cellular uptake of AuNPs is strongly size-dependent in the 10–100 nm range, with ~40–60 nm NPs typically exhibiting the highest cellular uptake efficiency in cancer cell lines [19,20,21]. For example, Huang et al. demonstrated that 45 nm was the optimal size for AuNPs (conjugated with ssDNA aptamers) to be internalized into HeLa cells, and it provided the highest efficiency for drug delivery. AuNPs of this size range are internalized via receptor-mediated endocytosis. Therefore, p-AuNP conjugates depend on endosome disruption capability to be delivered to the cytosol and to stimulate the cross-presentation of peptide T-cell epitopes through to the MHC class I processing pathway [22]. On the other hand, the surface properties of AuNPs are critical factors driving cellular uptake and transportation [23]. For example, the surface charge strongly affects uptake efficiency and endocytosis patterns of AuNPs, with several studies suggesting that AuNPs with positive charge are easier to be internalized into different cell lines [24].

Cell-penetrating peptides (CPPs) are short sequences of amino acids with the ability to transport cargo across the cell membrane. CPPs have been fused in tandem with T-cell epitopes to successfully deliver immunogenic peptides into antigen-presenting cells (APCs) [25,26]. In addition, this is a promising tool to improve intracellular delivery of nanomaterials [27,28]. However, the conjugation of CPPs to the NP surface is dependent on the properties of both partners, and further insights into this complex interplay are needed to improve the development of peptide nanoconjugates and to accelerate their applications in the vaccination field [29]. In addition, the extent of induced toxicity and internalization mechanisms remain a matter of debate, and further studies are required to improve their delivery to specific cell types with reduced side effects and enhanced efficacy [30,31,32].

The goal of this study was to explore in a systematic manner the role of relevant design variables of p-AuNP conjugates on the viability of THP-1 macrophages, cellular uptake, and endosomal cathepsin B activity in macrophages differentiated from the human monocytic leukemia cell line THP-1 (THP-1-derived macrophages) since this is a well-established cell model for in vitro evaluation of immune responses [33]. Therefore, AuNP concentration, AuNP size, peptide/CPP design, and the effect of the peptide density on the NP surface were investigated, providing insight into the design criteria and nanotoxicology effects associated with AuNPs and immunogenic peptides in this specific cell line (Figure 1).

## 2. Materials and Methods

### 2.1. Reagents

Tetrachloroauric (III) acid trihydrate (HAuCl_4_ 3H_2_O; 99.9%), trisodium citrate dehydrate, and sodium chloride were purchased from Merck (Darmstadt, Germany). All the solutions were prepared with ultrapure water (18.2 MΩ) obtained from a Millipore Simplicity Water System (Merck, Milford, MA, USA).

### 2.2. Peptides

The following peptides were synthesized by Shanghai Royo Biotech Co., Ltd. (Shanghai, China): p1 (CALNNKKPKYVKQNTLKLAT); p2 (CALNNKKPKYVKQNTLKLATRKKRQRRR); p3 (CALNNGPGPGPKYVKQNTLKLAT); and p4 (CALNNGPGPGPKYVKQNTLKLAT). Peptide sequences were confirmed by mass spectrometry, and purity was determined by reverse-phase HPLC (confirming 90–95% purity). Some experiments were carried out using p1 synthesized by standard solid-phase synthesis in a Rink amide resin (Iris Biotech GmbH, Marktredwitz, Germany) and 9-fluorenylmethoxycarbonyl (Fmoc)-protected amino acids, as described previously [34].

### 2.3. AuNPs Synthesis

For AuNPs of ~20 nm, the microsynthesis method previously reported by our group was used [17]. Briefly, a solution of 1 mM HAuCl_4_ (1 mL) was mixed with 100 μL of 40 mM trisodium citrate, adjusting the pH to 5.3 and incubating the mixture in a block heater at 105 °C for 30 min. The resulting AuNPs solution was cooled down to room temperature (RT), subsequently filtered with micropipette filter tips (polyethylene, 4 µm), and stored at 4 °C until used. AuNPs with a higher diameter were prepared by preheating the synthesis solutions at 60 °C, and 8 mL of 1 mM HAuCl_4_ was subsequently mixed with 800 μL of 10 mM trisodium citrate. Similarly, these larger AuNPs were heated for 30 min (105 °C), cooled down to RT, filtered, and stored until used.

### 2.4. AuNP/Peptide Surface Coverages

The reference AuNP:peptides molar ratio employed in this study was 1:100,000, which is a proportion reported by Egorova et al. that corresponds to a 100% surface coverage [35]. We opted to use it as a referential density of peptides coating the NP surface, which corresponds to 0.3 nmol-peptide/nm^2^ (from now on referred to as coverage 1.00). Two additional peptide surface densities were investigated by decreasing this ratio to 1:75,000 (coverage 0.75; 0.22 nmol-peptide/nm^2^) and increasing to 1:125,000 (coverage 1.25; 0.37 nmol-peptide/nm^2^), respectively. The functionalization of AuNPs of both sizes was carried out in a normalized way by maintaining equal peptide surface densities and thereby allowing the NP size to be studied as a distinct variable (Appendix A provides a detailed description and examples of the calculations). Briefly, peptides p1, p2, p3, and p4 were resuspended in ultrapure water (18.2 MΩ) at 30 mM, and variable volumes (12.6 to 179.9 μL) were mixed with 2 mL of the AuNPs solution (2.47 nM for Au23 and 0.054 nM for Au68) and 10% cleavage buffer (PBS 1X pH 8.8) to obtain the different surface coverages (0.75, 1.00, and 1.25). The solutions were stirred at 2000 rpm and 37 °C for 48 h in an orbital shaker (Labwit ZWY-103B, Shangai, China).

### 2.5. UV-Vis Spectroscopy

The concentration of AuNPs was determined spectrophotometrically with an Epoch^TM^ Microplate Spectrophotometer (BioTek Instruments, Winooski, VT, USA). A total of 100 μL of the AuNPs solutions (optical pathlength 0.3 cm) were poured in triplicate into a 96-well microplate, and the absorption spectra were recorded at wavelengths between 400–700 nm. The extinction coefficient (ε) was determined according to Liu et al. (2007) [36]. Gen5 3.00 software (BioTek) was used to collect the spectrophotometer output. Corrections to the AuNPs concentration were made following the microsynthesis methodology recently reported by our group to account for differences in gold mass associated with variations in NP size [17].

### 2.6. Zeta Potential

Zeta potential measurements were determined with a zeta potentiometer (Zetasizer ProBlue, Malvern Instruments, Westborough, MA, USA) at 25 °C and using a scattering angle of 90°. The Malvern Zetasizer software version 7.12 was employed to analyze the collected data. The measurements were averaged from 30 runs for each AuNP/coverage.

### 2.7. Electronic Microscopy

The core particle size and morphology of the AuNPs were determined by transmission electron microscopy (TEM) with 4 Å resolution (JEOL-JEM 1200EX-II, Tokyo, Japan) and a Gatan CCD camera for image acquisition (model 782; Gatan, Inc., Pleasanton, CA, USA). Frequency histograms were determined by processing the TEM images with ImageJ software version 1.8.0_201 (Fiji) [37].

### 2.8. Cell Culture

The human leukemia monocyte cell line THP-1 (ATCC TIB-202) was used to induce macrophage differentiation. THP-1 cells were cultured in RPMI 1640/L-glutamine medium supplemented with 10% fetal bovine serum (Biological Industries), 1% penicillin/streptomycin, 0.05 mM 2-mercaptoethanol (Merck), 1 mM sodium pyruvate (Gibco), 10 mM HEPES (Merck), and 13.9 mM glucose (Merck) and incubated at 37 °C in a humidified atmosphere of 5% CO_2_.

### 2.9. Differentiation of THP-1 Cells into M1 Macrophages

THP-1 monocytes were differentiated into M1 macrophages (from now on referred to as THP-1 macrophages) by adding phorbol 12-myristate 13-acetate (PMA; 200 ng/mL) to the culture medium and incubating for 24 h. THP-1 monocytes were subsequently stimulated with LPS (100 ng/mL) for 4 h to induce its polarization to M1 phenotype. Differentiation to macrophages was evaluated by flow cytometry.

### 2.10. Flow Cytometry

The phenotype of THP-1 macrophages was determined by measuring the expression of macrophage cell surface markers. For this purpose, 1 × 10^6^ THP-1 monocytes and macrophages were cultured, subsequently washed in phosphate buffer saline (PBS), and separated from the plate by treatment with Versene for 10 min at 37 °C. The labeling of surface molecules was carried out for 1 h at 4 °C, using the following antibodies: anti-human CD86 antibody-PE (374206, Biolegends Inc, San Diego, CA, USA), anti-human HLA DR antibody APC (307622, Biolegends, San Diego, CA, USA) for MHC class II molecules, anti-human CD40 antibody-FITC (E00279, BD, Franklin Lakes, NL, USA), and anti-CD11b APC antibodies (340937, Biolegends). Labeled cells were analyzed by LSR-Fortessa X20 flow cytometer (Becton Dickinson Biosciences, San Jose, CA, USA), and the collected data were processed using the FlowJo software (v10.10) (BD). The comparison was carried out between the cell population of THP-1 cells and THP-1 macrophages based on their size (forward scatter), granularity (side scatter), cell count (%), and mean fluorescence intensity (MFI).

### 2.11. Cell Viability Assays

#### 2.11.1. Cellular Metabolic Activity

THP-1 macrophages were seeded in 96-well plates at a density of 1 × 10^5^ cells per well and incubated with different nanoconjugates for 24 h at 37 °C in a 5% CO_2_ atmosphere. Cell viability was evaluated by measuring the metabolic activity of the cells by the MTT assay using the Vybrant^®^ MTT Cell proliferation assay kit protocol. Briefly, THP-1 cells were incubated with the MTT reagent (3-(4,5-dimethylthiazol-2-yl)-2,5-diphenyltetrazolium salt) upon incubation with the nanoconjugates, following the manufacturer’s instructions. The absorbance was then measured at a wavelength of 540 nm, using a multi-plate reader (Synergy 2, BioTek, Winooski, VT, USA). The percentage of metabolic activity was determined by normalizing with respect to the control (THP-1 macrophages without AuNPs).

#### 2.11.2. Cellular Death

THP-1 macrophages were seeded in 96-well plates at a density of 6 × 10^4^ cells per well and incubated with different nanoconjugates for 24 h at 37 °C in a 5% CO_2_ atmosphere. Then, cells were treated with the fluorescent probe Sytox Green (30 nM, Invitrogen), which was subsequently monitored for a period of 24 h using the IncuCyte^®^ S3 Live-Cell Analysis system (Sartorius GA, Göttingen, Germany). Real-time images of cells were acquired every 1 h and analyzed with IncuCyte^®^ S3 software to determine in each image the percentage of cell death based on the number of Sytox Green positive cells versus total SPY620-DNA positive cells.

### 2.12. Gold Labeling

The GoldEnhance™ LM gold labeling kit (Nanoprobes, Inc., Yaphank, NY, USA) was used according to the manufacturer’s instructions; it deposits additional gold onto the nanoconjugates internalized by the cells and enables their localization and counting under a light microscope. Briefly, THP-1 macrophages were seeded at 1.5 × 10^5^ cells per well in 24-well culture plates with 12 mm round glass cover subjected to UV light beforehand. Later, AuNPs were added to each well and incubated for 24 h at 37 °C in a 5% CO_2_ atmosphere. The supernatant was carefully removed, and the wells were washed with three buffers: PBS 1X pH 7.6, PBS 1X with gelatin 1%, Tween-20 0.05%, and NaCl 0.5 M pH 7.6; finally, the supernatant was washed with nanopure water. Round glass covers were then removed carefully and placed on a slide. A total of 100 μL of the gold enhancement solution was added (20 min), and the reaction was stopped with nanopure water. Samples were finally observed under a light microscope by taking images at 100× magnification with immersion oil. The images were analyzed with ImageJ software, and the percentage of positives was quantified with respect to control (THP-1 macrophages without AuNPs).

### 2.13. Cathepsin B Activity Assay

THP-1 macrophages were seeded in 96-well plates at a density of 1 × 10^5^ cells per well and incubated with different conjugates for 24 h at 37 °C in a 5% CO_2_ atmosphere. The cathepsin B activity assay kit (ab65300; Abcam, Cambridge, UK) was employed to determine cathepsin B activity. Briefly, THP-1 macrophages were washed with 1X PBS and homogenized with 50 μL of lysis buffer. Then, whole cell lysate was added in dark 96-well plates, together with 50 μL reaction buffer and 1 μL of cathepsin B substrate Ac-RR-AFC (amino-4-trifluoromethyl coumarin, 10 mM). The plates were incubated at 37 °C for 2 h, protected from light, and the fluorescence of the cathepsin B-cleaved substrate was measured at 400/505 nm excitation/emission with a fluorescence microplate reader (Synergy H1, BioTek, Winooski, VT, USA). Free peptides were used as a positive control (10 µM, p1–p4), and non-conjugated (peptide-free) AuNPs (23 and 68 nm) were used as negative controls.

### 2.14. Cathepsin B Cleavage Prediction

Putative cleavage sites for cathepsin B were calculated by implementing the algorithm described by Ferrall-Fairbanks et al. [38]. In brief, a sliding window of eight residues in length assessed the cleavage likelihood using residue scores of a cathepsin B specificity matrix obtained from the MEROPS database [39]. This matrix quantifies the propensity of amino acids in positions P4–P4′ to promote cleavage at the P1–P1′ bond. Cumulative scores for each window were calculated by summing up individual residue scores, with higher values indicating a greater substrate specificity for the protease’s active site. Cleavage sites were ranked according to their scores, and the top candidates were visualized on annotated protein sequences using the Python library DNA Features Viewer, with arrows highlighting the most prominent cleavage positions.

### 2.15. Statistical Analysis

Quantitative results are shown as mean ± standard deviation (SD). The Shapiro–Wilk test was applied to assess normal distribution, while the parametric or nonparametric *t*-test was employed for comparison between the two groups. For three or more groups, ANOVA with post-hoc Sidak analysis was used. Nonlinear analysis was applied to determine the IC_50_ value. A *p*-value (*p*) < 0.05 was considered statistically significant. GraphPad Prism 10.3. 0 was employed for statistical analysis and for creating the graphs.

## 3. Results

### 3.1. Physicochemical Characterization of AuNP Nanoconjugates

The four peptide sequences investigated in this study start at the N-terminal site by the CALNN pentapeptide, which anchors the peptide to the gold surface by forming a covalent bond with the thiol group of the cysteine (Table 1). This short region allows the formation of a dense self-assembled monolayer on the AuNPs, enabling extremely stable and water-soluble nanoconjugates [40,41]. The core region in all the peptides is the promiscuous (“universal”) CD4+ T-cell epitope HA_306−318_ (PKYVKQNTLKLAT) from influenza A virus [42,43], which was chosen as a model T-cell epitope for peptide design purposes. Indeed, the broad MHC class II specificity of this peptide can be analyzed from a population coverage point of view using the Predivac-3.0 (epitope discovery mode) method previously developed by our group [7]. As an example, this tool predicts a nonameric core on peptide 1 (YVKQNTLKL) that would potentially cover 93.15% of the Chilean population. (Appendix A lists predicted specificity for MHC class II alleles.) Predivac-3.0 performs in silico T-cell epitope mapping by implementing a scoring method (SDR approach) that considers specific MHC class II alleles present in the target population and subsequently calculates the theoretical population coverage based on allele frequencies available at the Allele Frequency Net Database (AFND) [44].

In this study, we focused on comparing the effect of two different epitope flanking regions (KK and GPGPG) that play an important role in minimizing junctional immunogenicity and preserving the identity of each individual epitope during the processing of vaccine constructs [45]. The KK linker has often been used to join CD4+ T-cell epitopes and/or B-cell epitopes, as this is a target motif for the cathepsin B in the endosomal-lysosomal system associated with MHC class II-mediated antigen processing [46,47]. On the other hand, the glycine-rich GPGPG linker has been shown to confer structural flexibility to epitope-based vaccines, disrupting junctional epitope formation while enhancing immune processing and epitope presentation [48].

Figure 2 presents TEM images and zeta potential measurements of peptide-free AuNPs and p2-AuNPs, showing a high monodispersity and spherical sizes in both cases (see details in Appendix A). The resulting AuNPs under the two synthesis conditions presented average diameters of 23.4 ± 2.5 nm (from now on referred to as Au23; Figure 2A) and 68.2 ± 8.3 nm (from now on referred to as Au68; Figure 2D). We employed the nomenclature p-Au23 and p-Au68 to refer, respectively, to Au23 and Au68 conjugated with each peptide (where p is p1, p2, p3, or p4).

As expected, slight differences in sizes were determined between AuNPs nanoconjugates and free AuNPs, reaching diameters of 25.4 ± 1.9 nm for p2-Au23 (Figure 2B) and 64.0 ± 16.3 nm for p2-Au28 (Figure 2D). In addition, the ζ-potential (pZ) of the nanoconjugates accounts for differences in the surface charge of the NPs that are attributed to the peptide layer on the AuNPs surface (Figure 2C,F). Accordingly, the pZ value of p2-Au23 (30.62 ± 0.37 mV) was more positive than that of non-functionalized Au23 (−42.64 ± 0.71 mV), while for p2-Au68 and Au68, the resulting pZ values were, respectively, 38.77 ± 0.75 mV and −40.45 ± 1.83 mV (Figure 2F). These results are consistent with the amino acid composition of peptide p2, which possesses five arginine residues and seven lysine residues, resulting in an increase in the positive charge on the surface of AuNPs (Appendix A). In addition, the absorption spectra of both Au23 and Au68 showed typical curves with a peak at 520 nm associated with the surface plasmon resonance (SPR) band both for the peptide-free AuNPs and the nanoconjugates (Appendix A).

Figure 3 shows a comparison between p-Au23 (Figure 3A) and p-Au68 (Figure 3B) at equal peptide surface densities. Peptide coating provided positive surface charges to the nanoconjugates in both cases (up to 35–40 mV), which accounts for successful peptide conjugation by analogy with the negative pZ of free AuNPs (−40 mV). However, slightly higher pZ values were determined (coverage 1.25) for p-Au68 (30.73–38.77 mV) in comparison with p-Au23 (20.02–34.21 mV). The closeness of both pZ ranges is consistent with the methodology employed to normalize the surface density at peptide coverages 0.75 (0.22 nmol-peptide/nm^2^), 1.00 (0.30 nmol-peptide/nm^2^), and 1.25 (0.37 nmol-peptide/nm^2^). Figure 3A shows slight increases in the pZ values associated with higher surface coverages of p-Au23, suggesting that the curve is approaching a plateau where the maximum pZ for each peptide is potentially reached. This behavior is consistent with the molar ratio (AuNP:peptides) reported by Egorova et al. (1:100,000) to achieve full peptide surface coverage, even though for p-Au68 the nanoparticle surface would become saturated with positive charges at a lesser ratio of 1:75,000 (coverage 0.75) [35]. On the other hand, only for p-Au23, a variation on pZ as a function of the peptide surface coverages is verifiable. By contrast, the plateau in the pZ curve for p-Au68 indicates that AuNP surface is saturated with positively charged residues (p1 to p4).

AuNPs conjugated with peptides p2 and p4 exhibited the highest positive charge in both NP sizes (Au23 and Au68), which is consistent with the amino acidic composition of these peptides and the presence of an arginine-rich cell-penetrating peptide (CPP) situated at the distal end. Indeed, p2 and p4 contain 13 and 11 basic residues, respectively, while p1 and p3 contain only 6 and 4 positively charged residues (Appendix A). This CPP is derived from the transactivator of transcription (pTAT_48–60_) of human immunodeficiency virus (HIV), a sequence that has been widely studied in cargo delivery applications thanks to its ability to translocate across cell membranes without compromising their integrity [49].

### 3.2. Differentiation of THP-1 Monocytes to Macrophages

THP-1 monocytes were differentiated from macrophages using PMA and LPS protocols, as indicated in Section 2.9. THP-1 macrophages were characterized by flow cytometry through an increase in forward (Appendix A) and side scatter (Appendix A), with all the cells resulting positive for CD11b (~98%, Appendix A). Moreover, ~20% of the cells were positive for CD40 (Appendix A), ~99% for HLA-DR (Appendix A), and ~15% for CD86 (Appendix A). Interestingly, the levels of HLA-DR increased (2.1-fold) in macrophages compared to monocytes (Appendix A). However, the protein levels of CD40 (Appendix A) and CD86 (Appendix A) did not change.

### 3.3. Effects of p1-AuNPs Concentration on Cell Viability

The effect of p1-AuNPs upon cell viability was evaluated on THP-1 macrophages, whose differentiation was confirmed from the increment in forward and side scatter, CD11b expression, and HLA-DR MFI (Appendix A). Figure 4A shows that peptide-free Au23 reduced the cell viability of THP-1 macrophages at ~12% at a 1000 pM concentration, whereas p1-Au23 resulted in a significant cell viability reduction at 100 pM (~11%) and 1000 pM (~22%) (Figure 4B). In addition, p1-Au23 showed a viability reduction of ~10% in comparison with peptide-free Au23 (Figure 4C). Given this, we selected 10 pM of AuNPs as the working concentration for the next experiments. Inhibitory concentrations (IC_50_) were calculated for peptide-free Au23 (7280 ± 1720 pM) and p1-Au23 (3308 ± 610 pM) (Figure 4D). The difference between IC_50_ values was statistically significant, proving that concentrations higher than 1000 pM are toxic for this cell line. Similar effects were obtained using Sytox Green in the Incucyte system (Appendix A).

### 3.4. Effects of Size and Coverage of AuNPs on Cell Viability

As presented in Figure 4, THP-1 macrophages exposed to p-Au23 maintained their metabolic status (~100% cell viability). However, treatment with p-Au68 nanoconjugates reduced their metabolic activity for all the conditions (Figure 5A–D). This effect resulted to be higher for p2-Au68 (reaching ~20–50% viability) and p4-Au68 (reaching ~50% viability) than that of Au68 conjugated with p1 and p3 (reaching ~60–70% viability). In addition, the metabolic activity was drastically reduced to a similar level to that of the positive control for p2-Au68 nanonconjugate (Figure 5B).

### 3.5. Effects of Size and Coverage of AuNPs on Cellular Uptake

Figure 6 shows that THP-1 macrophages exposed to p-Au23 presented non-significant cell death in a consistent manner with the metabolic activity measurements. By contrast, treatment with p-Au68 increased the percentage of cellular death under all conditions tested (Figure 6A–D). However, the difference in cell death effects caused by incubation with p1-Au68 (~8%) and p3-Au68 (~22%) is considerably higher in comparison with exposing the cells to p2-Au68 (~40%) and p4-Au68 (~46%). Furthermore, it is worth noting that p4-Au68 with coverage 0.75 showed the lowest cellular death (36.6%) with respect to coverages 1.00 (61.4%) and 1.25 (59.3%). In these two conditions, cell death reached similar percentages than that of the DMSO control (Figure 6D).

Figure 7 presents the uptake results of p-Au23 nanoconjugates by THP-1 macrophages, which was evaluated for peptides p1, p2, p3, and p4. The cellular uptake for p-Au23 nanoconjugates was higher for all peptides evaluated in this study, which is visualized by black dots in the bright field compared to the control (Figure 7). Peptide p1-Au23 showed significant uptake with the three coverages (0.75, 1.00, and 1.25), which suggests an increasing trend at higher coating densities (Figure 7B). Treatments with peptides p2 and p3 showed significant uptake only for p-Au23 at coverage 0.75 (Figure 7C,D), while p4-Au23 presented cellular uptakes at all conditions (Figure 7E). However, cellular uptake of p-Au68 nanoconjugates did not show a variable pattern. Indeed, p1-Au68, p3-Au68, and p4-Au68 showed a reduction compared with the control cells. Only p2-Au68 (coverage 1.25) showed a reduction when compared with other conditions.

### 3.6. Effects of Size and Coverage of AuNPs on THP-1 Macrophage Cathepsin B Activity

Bioinformatics prediction of cathepsin B cleavage sites in the four peptide sequences shows that KK and GPGPG are the preferred motifs, which would potentially favor the delivery of the peptides from the AuNPs (Appendix A). Some cleavage sites additionally predicted by the implemented algorithm are situated within the CD4+ T-cell epitope region, which could potentially disrupt subsequent processing and presentation of the epitope in the context of MHC class II proteins. However, these are probabilistic events, and variable sequences are potentially produced by the enzymatic activity in the endosomal compartment (Table 2).

The four putative sequences are predicted to release the nonameric core of the T-cell epitope (YVKQNTLKL). The cleavage patterns are slightly different for peptides differing in the presence of the pTAT region at the C-terminus because this segment modifies the predictive scores as it alters the top-ranked scored sites.

Figure 8 presents normalized cathepsin B activities resulting from the treatments of THP-1 macrophages with Au23 and Au68 conjugated with peptides p1 (Figure 8A), p2 (Figure 8B), p3 (Figure 8C), and p4 (Figure 8D), while raw cathepsin B activities are presented in Appendix A.

Incubation with p1-AuNPs (23 and 68 nm) and p3-AuNPs (23 and 68 nm) nanoconjugates resulted in moderate levels of enzymatic activity, with mostly no significant differences compared to the free medium (basal activity; black bars) and slightly lesser activities than that of the corresponding free peptides (positive controls; grey bars). However, p2-Au23 and p4-Au23 presented high levels of cathepsin activation, with statically significant differences for p2-Au23 at *p*-value < 0.001 (coverages 0.75 and 1.00) and *p*-value < 0.0001 (coverage 1.25) and for p4-Au23 at *p*-value < 0.95 (coverages 1.00 and 1.25). The highest normalized activity was determined for p4-Au68 (coverage 1.00), which surpassed the level of 300 AU.

The increased cathepsin B activity observed in Figure 8D (p4-Au68 1.00) may result from a complex dynamic interplay between factors such as the resulting redox environment and pH changes generated by the internalized nanoconjugate. Previous work has shown that high levels of AuNPs can inhibit cathepsin B activity due to increased intracellular levels of ROS [50], which is the main mechanism driving gold nanoparticle toxicity and its oxidative effect on cysteine catalytic residues [51]. However, the correlation between ROS and cathepsins extends beyond a simple cause-and-effect relationship in regulated cell death pathways such as apoptosis and autophagy [52]. While ROS promotes apoptosis by inducing the release of cathepsins into the cytosol through disruption of lysosomal integrity, recent studies have suggested this process may also influence the subcellular localization of cathepsins [53]. For instance, cathepsins can translocate into the nucleus, where their activity appears to be enhanced during apoptosis [54,55]. In another study, Lagadic-Gossmann et al. observed changes in the expression and activity of cathepsins B and D in the rat pheochromocytoma cell line PC12 upon treatment with H_2_O_2_. Since cathepsins are known to activate at acidic pH levels within lysosomes, the authors of this study hypothesized that these proteases might also become active in the acidified cytoplasm during H_2_O_2_ and nitric oxide-induced apoptosis [56]. However, further investigations are necessary to unveil the mechanisms underlying these phenomena.

Interestingly, Figure 9 shows that cathepsin B activity exhibits a tendency toward greater dispersion levels as a function of pZ when considering the normalized activity for all AuNPs (p-Au23 and p-Au68). The correlation does not show a significant change (Y = 0.3826 × X + 155.7; r^2^: 0.0019; P: 0.8369). The normalized activity values fluctuated around an average of 157.1 UA, clustering into three main groups with increasing standard deviations: (i) 155.7 ± 24.01 UA (pZ around 20 mV); (ii) 149.7 ± 37.06 (pZ around 30 mV); and (iii) 168.8 ± 107.2 (pZ around 40 mV).

## 4. Discussion

It is well-established that AuNPs accumulate in macrophages due to their strong phagocytic capacity, making these cells suitable for investigating the toxic effects of p-AuNPs nanoconjugates [57,58]. However, the functional impact of AuNPs exposure on immune system cells remains poorly documented, even though the uptake of large quantities of these NPs may severely alter cellular metabolism [59,60]. THP-1 macrophages are a well-described model for in vitro evaluation of immune responses [33,61], allowing to improve the understanding of key nanoconjugate design parameters and their effects on cellular function. Achieving such a level of control is difficult in vivo since factors such as biodistribution, immune system variability, and tissue-specific responses introduce additional complexity. While in vitro nanotoxicity findings may not always reproduce in vivo outcomes, these results shed light on the mechanisms involved in nanoconjugate uptake and toxicity at the cellular level. Furthermore, they are useful to guide future in vivo studies focused on the safety and effectiveness of AuNPs as a nanocarrier and delivery system for immunogenic peptides [62]. However, further research in animal models is required in order to consider the physiological environment and variables that could significantly alter the behavior of p-AuNPs, including serum protein interactions, immune modulation, and biodistribution [63].

The use of AuNPs to deliver peptide antigens into the immune cells is considered a promising approach to overcome the low immunogenicity of synthetic peptides and to develop advanced targeting strategies [64]. The size and surface properties of AuNPs are crucial factors that drive immunological effects and influence the thermodynamic parameters governing uptake efficiency and endocytic patterns [65]. However, the literature is far from consistent and peptide conjugation of AuNPs can be toxic for cells or even restrict their penetration across the plasma membrane [66].

An optimal range of AuNPs sizes (~40–60 nm) has been confirmed in studies carried out in a cancer cell lines (e.g., HeLa cells). For example, Wang et al. investigated the uptake by HeLa cells of AuNPs with diameters of 13, 45, 70, and 110 nm [65]. This study proved that 45 nm AuNPs entered the cells via endocytosis, showing that 61% of the NPs internalized and accumulated inside endocytic vesicles. However, cellular uptake decreased as the particle size increased, with larger AuNPs remaining mostly bound to the cell membrane (23% uptake for 70 nm AuNPs and no uptake for 110 nm AuNPs). Similarly, Sun et al. examined the translocation of peptide-capped AuNPs of three different sizes (13, 30, and 60 nm) into HeLa cells. Their findings indicated that increasing the NP diameter to 60 nm, while maintaining a constant surface density of peptide ligands, resulted in reduced cellular internalization [67].

We opted to compare a standard size produced by a Turkevich-based method (~20 nm) with a size (~70 nm) that is slightly above the range typically associated with top-performing cellular uptake of AuNPs. Three peptide surface coverages were investigated (coverages 0.75, 1.00, and 1.25), while attempts to study the effect of lesser surface densities were made difficult due to the low stability of the resulting nanoconjugates. In these cases, pZ values were closer to electroneutrality, and AuNPs tended to aggregate. As peptide conjugation involves an exchange of citrate anions on the AuNP surface, their covalent binding to the Au^0^ surface produced stable dispersion when pZ variations fluctuated from −40 mV (citrate coating AuNPs) to at least +14 mV (peptide-coated AuNPs). This represents a methodology challenge derived from the charge properties of the system, which requires further optimization of the experimental conditions. However, previous observations suggest that NPs with homogeneously charged surfaces may experience abrupt precipitation when approaching charge neutrality [68]. Indeed, it is also possible that peptides on p2-AuNPs and p4-AuNPs adopt a tighter arrangement and present a more uniform distribution of positive charges due to the positively charged pTAT region that extends outward from the AuNP surface.

The high internalization observed for p1-Au23 (coverage 1.25), p2-Au23, and p4-Au23 (Figure 8) is consistent with cathepsin B activity measurements for p-Au23 nanoconjugates. Figure 9 reveals that the normalized cathepsin B activity (enzymatic response divided by cell death) resulting from p-Au68 treatment is increasingly variable as the pZ value increases. This is especially noticeable for the most positively charged nanoconjugates (pZ ~40 mV), which strongly enhances or impairs the enzymatic activity (168.8 ± 107.2). By contrast, the treatment with p-Au23 nanoconjugates allows to maintain stable protein levels of the enzyme at pZ values around 20 mV (155.7 ± 24.01 AU), with a tendency to increase cathepsin B activity at pZ nearby 30 mV. In particular, Figure 8 confirms that peptides p2 and p4 exhibit excellent properties in terms of cathepsin B activation, demonstrating statistically significant increases across most surface coverages in p2-Au23 (0.75, 1.00, and 1.25) and p4-Au23 (1.00 and 1.25). In addition, the similar cathepsin B activities observed for these two nanoconjugates (p2-Au23 and p4-Au23) and the corresponding free peptides (control) is consistent with endocytosis as a primary mechanism of cellular entry for both CPP-containing peptides and the nanoconjugates. This result further demonstrates that AuNPs can effectively internalize a comparable amount of peptides into the endosomal/lysosomal compartment of THP-1 macrophages, addressing the weaknesses of synthetic peptides, which are inherently unstable, have short half-lives, and undergo rapid in vivo elimination [69]. It is worth noting that p2-Au68 (coverage 1.00) presented the highest cathepsin B activity; however, this AuNP size caused deleterious effects on cellular viability of THP-1 macrophages both in terms of mitochondrial activity (Figure 5) and cell death (Figure 6).

The bioinformatics design of the peptide sequence with cathepsin B-cleavable motifs was successfully achieved, based on the underlying assumption that this element can facilitate the correct release of the T-cell epitope for further processing through the MHC class II antigen presentation pathway [70,71]. Cleavage sites were predicted in the flanking region between the motifs (spacers) and the CALNN linker attached to the gold nanocarrier (Appendix A). Additional cleavage events predicted within the T-cell epitope region could potentially disrupt the nonameric binding core (YVKQNTLKL); however, these are probabilistic events, and variable sequences are expected to be produced for subsequent antigen processing pathways (Table 2). The model CD4+ T-cell epitope provides an example of promiscuous binding to a high number of MHC class II proteins described in the Chilean population, which would potentially account for 93.15% coverage according to the Predivac-3.0 immunoinformatics tool. The rational peptide design aims here to optimize the inclusion of T-cell epitopes (viral pathogens), maximizing the coverage in a given target population [7,72].

The cellular viability of THP-1 macrophages exposed to free Au23 and p1-Au23 was slightly reduced at a concentration of 100 pM, leading us to select a working range between 1–10 pM. Previous studies in murine monocytes (RAM.7 and J774A1 cell lines) and human macrophages (K562 cell line) described toxicity effects at AuNPs concentrations of ~1000 pM [73,74,75]. Our findings of severe toxicity of p-Au68 in THP-1 macrophages are, to some extent, unexpected in light of the commonly held assumption that short NPs are generally more damaging to the cell integrity due to enhanced penetration capabilities [76]. For example, a recent study assessed the toxicity of polyethyleneimine (PEI) and polyethylene glycol (PEG) functionalized AuNPs with sizes similar to those employed herein (20 and 50 nm), finding both to be safe (non-toxic) and having high cellular uptake of the nanoconjugates [77].

The cytotoxicity of p-Au68 was particularly high when conjugated with peptides p2 (~30–50% cell death) and p4 (~40–60% cell death). These results are consistent with the observed reduction in metabolic activity derived from treatments with p2-Au68 (~40–50% decrease) and p4-Au68 (~40–60% decrease), while Au68 conjugated with p1 and p3 showed a lower reduction in cellular viability (~10–20% cell death). These peptides differ from each other by the presence of the arginine-rich TAT peptide (YGRKKRRQRRR). It is noteworthy that pTAT-functionalized NPs are considered a promising approach to overcome cell-membrane barriers in a non-invasive manner [78,79], while cellular uptake has been enhanced with CPP-translocated cargoes with a net positive charge [80]. For example, Saar et al. investigated the membrane toxicity of five CPPs and showed that pTAT(48–60) was completely innocuous to K562 human erythroleukemia cells [81]. Furthermore, cell-specific mechanisms for cellular entry of pTAT can be inferred from the poor or negligible permeability in some cell lines [82]. In addition, pTAT has been reported to produce low toxicity in three cell lines (HeLa, A549, and CHO) but becoming toxic at concentrations above 100 μM when conjugated to a peptide derived from the NEMO-binding domain (NBD) [83].

The effect on cell death caused by p-Au68 could be partially associated with a direct effect on the cell membrane integrity, which is further supported by the damaged state of the cells observed by microscopy (Figure 7F–J). Previous works have also found that AuNPs functionalized with cationic head groups can cause membrane disruption at an increased AuNPs concentration by embedding themselves within the lipid bilayer and destabilizing the entire membrane structure [84]. Another study addressing the interaction of positive charge-modified AuNPs with phospholipid membranes observed the formation of transmembrane pores, with a diameter of 3.0 nm, which resulted in the destruction of the cell membrane function [85]. However, additional mechanisms involved in cell death must be considered. In particular, mitochondrial dysfunction (Figure 5) is likely a consequence of AuNP-induced oxidative stress, which additionally amplifies oxidative damage and toxicological responses, such as inflammation and viability loss [86,87]. For example, a previous work has described the accumulation of positively charged AuNPs in the outer portion of the mitochondrial membrane and the consequent cell death due to leakage of apoptotic proteins (p53 and caspase-3) into the cytosol [88]. Given the well-established intracellular effects of electropositive AuNPs, it was interesting to find that cathepsin B activity presented a strong negative correlation with pZ values. The association between these variables might account for the growing capability of p-AuNPs to be internalized and directed to the endosomal compartment.

The two nanoconjugates (p-Au23 and p-Au68) were synthesized with equal peptide surface coverages, which would explain why AuNPs conjugated with peptide p4 (p4-Au23 and p4-Au68) exhibited similar pZ values. Importantly, this parameter is directly associated with the surface charge density (i.e., the amount of electric charges per surface area) [89]. However, the pZ curves for p1, p2, and p3 showed higher values for p-Au68 in comparison with p-Au23. As far as our understanding extends, the electrophoretic mobility, which is the underlying physical principle of the pZ measurement, is expected to be higher for larger NPs [90,91]. Therefore, this phenomenon may partially explain the increased pZ values observed for Au68, which has a volume 26 times larger than that of Au23. Furthermore, the sensitivity of this technique could be potentially limited for greater NPs, making it more difficult to discern subtle variations in surface charge associated with different peptide coatings. In line with this reasoning, although pZ has been successfully used to characterize the net charge of NPs and to predict NP toxicity [92], a correlation with cytotoxicity is not always observed [88]. For example, Weiss et al. recently demonstrated that electrokinetic charge (Qek; µmol/g) rather than the absolute value of the pZ could be a better descriptor of toxicity, which was carried out by comparing five cationic carbon nanotubes exhibiting a similar pZ value (from + 20.6 to + 26.9 mV) but displaying an increasing electrokinetic charge (from 0.23 to 4.39 µmol/g) [93].

Therefore, the toxic effects delivered by p-Au68 nanoconjugates to THP-1 macrophages can be linked with their high number of positive charges on the NP surface (in comparison with smaller p-Au23). On the other hand, an additional effect of the larger size is the higher amount of gold mass at equal nanoparticle concentrations. The gold-atom content of p-Au68 is approximately 26 times higher per particle, while the surface area of p-Au68 is 9 times larger than that of p-Au23 particles (Appendix A). Given this, the massive cellular damage evidenced by cellular debris observed in Figure 7F (in comparison with Figure 7A) might be associated with the high gold content in p-Au68 nanoconjugates. The consequent release of cytotoxic Au^+^ ions is expected to increase the cellular oxidative stress and induce mitochondrial dysfunction, potentially leading to necrosis activation. These effects are triggered by disruption of the mitochondrial membrane integrity, triggering the release of cytochrome C and inducing caspase-dependent apoptosis [63].

## 5. Conclusions

Overall, this study shows that peptide-conjugated AuNPs of two sizes (23 and 68 nm) delivered contrasting effects in the viability of THP-1 macrophages, with the latter nanoconjugate (p-Au68) causing severe toxicity in terms of mitochondrial functionality and resulting cell death. Possible mechanisms involve effects over the cell membrane integrity caused by the greater number of positive charges on the larger AuNP, as well as pronounced oxidative damage and necrosis caused by the high amount of gold mass potentially activating the apoptotic machinery. The pZ curves of the nanoconjugates revealed that the NP surfaces were predominantly saturated with positively charged peptides. Although the differences in peptide surface coverages were minor, in some cases, they were sufficient to induce statistically significant effects in the biological assays. Consistent with this, the CPP region (pTAT) included in two of the peptides (p2 and p4) strongly decreased the viability of THP-1 macrophages when conjugated with p-Au68 but did not cause negative effects with p-Au23. In addition, the p2-Au23 and p4-Au23 nanoconjugates demonstrated excellent properties for transporting peptides into the macrophages cell line, which was further supported by high levels of cathepsin B activity. These findings provide valuable insight into the design criteria of AuNPs and immunogenic peptides, underscoring the relevance of NP size and surface charge to balance delivery efficacy with nanotoxicity effects in the context of peptide-based nanovaccination development.

## Figures and Tables

**Figure 1 vaccines-13-00119-f001:**
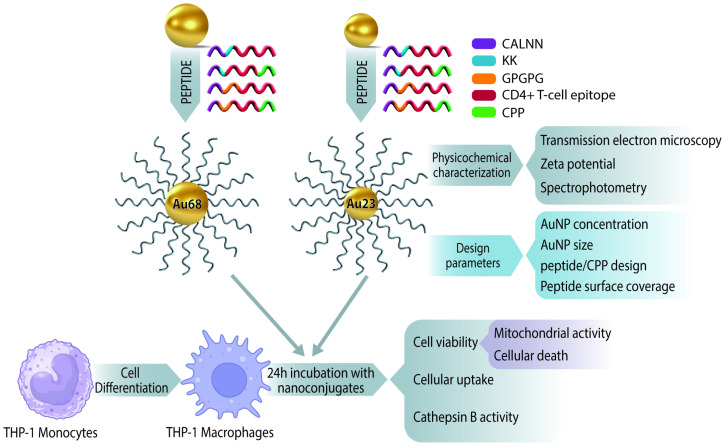
Schematic illustration showing the experimental strategy followed in the research to evaluate the effect of AuNP concentration, AuNP size, peptide/CPP design, and peptide density on the cellular function of THP-1 macrophages (cell viability, cellular uptake, and cathepsin B activity).

**Figure 2 vaccines-13-00119-f002:**
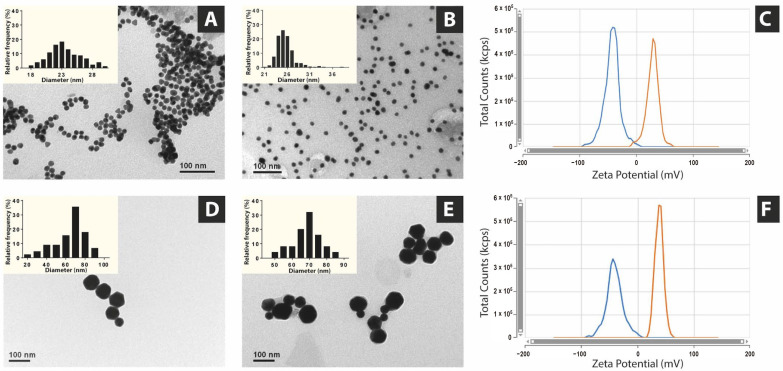
TEM micrographs of (**A**) peptide-free Au23, (**B**) p2-Au23 conjugates, (**D**) peptide-free Au68, and (**E**) p2-Au68 conjugates, showing the particle size distributions (histogram insets). Zeta potential distributions are presented in (**C**) for Au23 (blue line) and p2-Au23 (orange line) and in (**F**) for Au68 (blue line) and p2-Au68 (orange line).

**Figure 3 vaccines-13-00119-f003:**
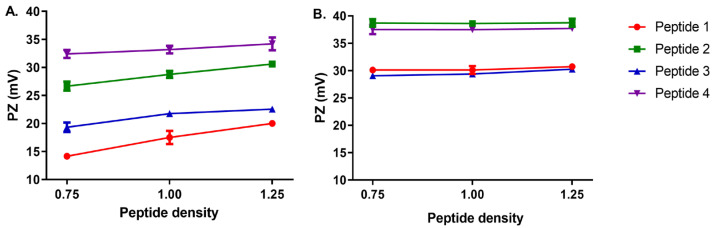
Effect of peptide surface coverage on the surface charge of (**A**) Au23 and (**B**) Au68 conjugated with p1 (red), p2 (green), p3 (blue), and p4 (purple).

**Figure 4 vaccines-13-00119-f004:**
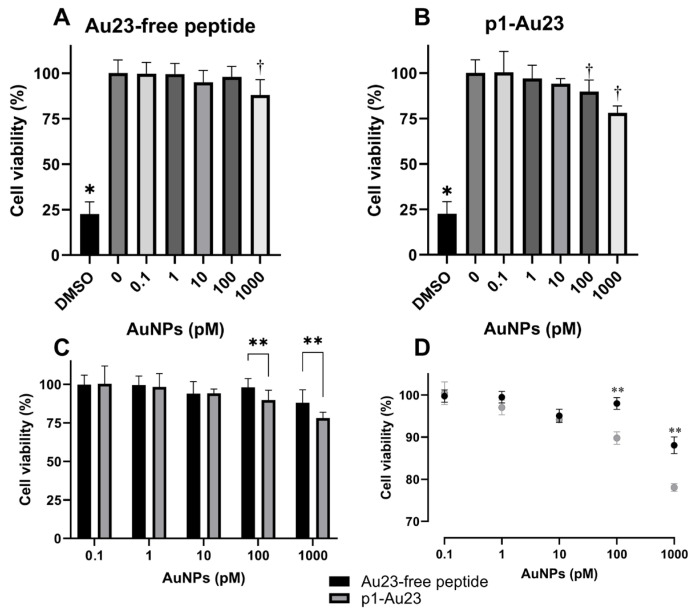
Effect of AuNP concentrations on the cell viability of THP-1 macrophages. Metabolic activity was evaluated by MTT assays upon 24 h incubation of the cell line with (**A**) peptide-free Au23, (**B**) p1-Au23, (**C**) comparison between Au23 (black) and p1-Au23 nanoconjugate (grey), and (**D**) IC_50_ determination for Au23 (black) and p1-Au23 nanoconjugate (grey). Cells were treated with 6% dimethyl sulfoxide (DMSO) as a positive control for low metabolic status. * *p* < 0.05 with respect to all conditions for peptide-free Au23 and p1-Au23. ^†^*p* < 0.05 with respect to the concentrations. ** *p* < 0.05 between both conditions (n = 5).

**Figure 5 vaccines-13-00119-f005:**
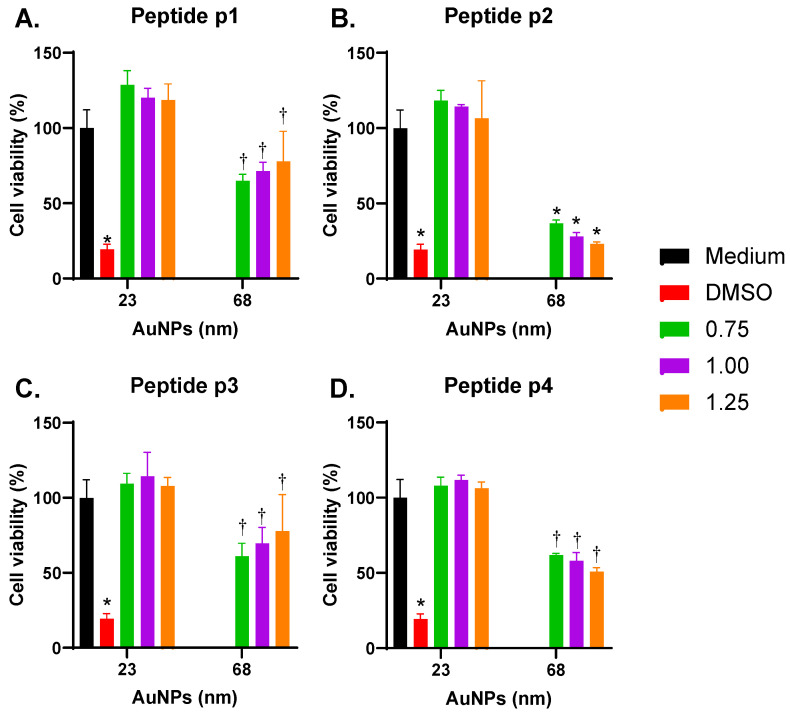
Effect of the AuNPs size and peptide surface coverage on cell viability of THP-1 macrophages. Metabolic activity was evaluated with MTT assay upon 24 h incubation of the cell line with each nanoconjugate of 23 and 68 nm: (**A**) p1-AuNPs, (**B**) p2-AuNPs, (**C**) p3-AuNPs, and (**D**) p4-AuNPs. Cells were treated with 6% DMSO as a positive control for low metabolic status. * *p* < 0.05 with respect to all conditions for p1, p3, and p4, for p2 with respect to the culture medium, and Au23 (0.75, 1.00, and 1.25). ^†^
*p* < 0.05 for Au68 with respect to the same condition in Au23.

**Figure 6 vaccines-13-00119-f006:**
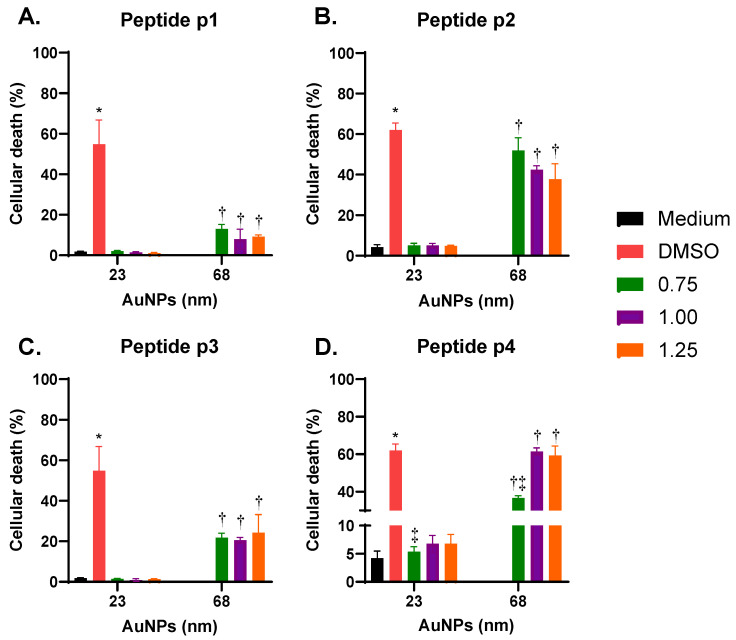
Effect of p-AuNPs on cellular death in THP-1 macrophages. THP-1 macrophages were incubated in the presence of Sytox Green with AuNPs conjugated, at different surface densities, with (**A**) peptide p1; (**B**) peptide p2; (**C**) peptide p3 and (**D**) peptide p4. The fluorescent probe was quantified by Incucyte. 6% DMSO was utilized as a positive control of cellular death. * *p* < 0.05 with respect to all conditions. ^†^ *p* < 0.05 with respect to the same condition in 23 nm. ^‡^ *p* < 0.05 with respect to 1.00 and 1.25 of coverage at the same size.

**Figure 7 vaccines-13-00119-f007:**
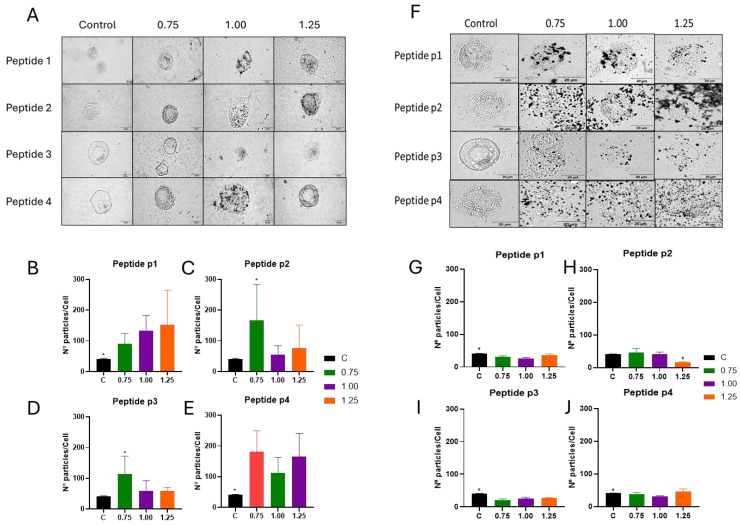
Cellular uptake of p-Au23 and p-Au68 nanoconjugates by THP-1 macrophages, considering the four peptides (p1, p2, p3, and p4) and surface coverages of 0.75, 1.0, and 1.25. Light microscope photographs (100× magnification) of gelatin-fixed THP-1 macrophages incubated for 24 h with (**A**) p-Au23 and (**F**) p-Au68 nanoconjugates, showing AuNPs using the gold enhance labeling assays (black dots). The count of p-Au23 nanoconjugates in THP-1 cells is presented for peptides (**B**) p1, (**C**) p2, (**D**) p3, and (**E**) p4. The count of p-Au68 nanoconjugates in THP-1 cells is presented for peptides p1 (**G**), p2 (**H**), p3 (**I**), and p4 (**J**). The control corresponds to the treatment with p-AuNPs-free culture medium. * *p* < 0.05 denotes statistically significant differences between two conditions (n = 3).

**Figure 8 vaccines-13-00119-f008:**
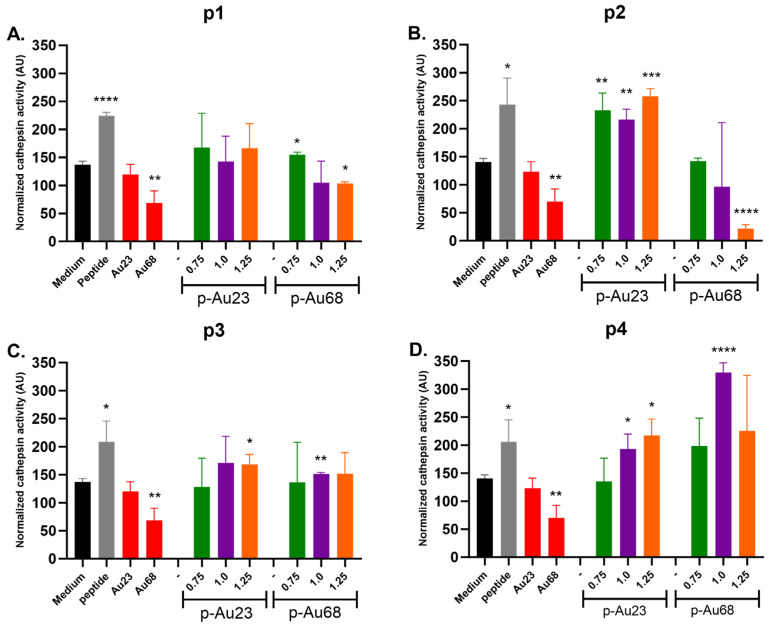
Normalized cathepsin B activity measured in THP-1 macrophages incubated with AuNPs conjugated with peptides (**A**) p1, (**B**) p2, (**C**) p3, and (**D**) p4 at different surface coverages (0.75, 1.00, and 1.25). The normalization corresponds to the rate between cathepsin B activity and the percentage of cellular survival (in arbitrary units). Controls correspond to free peptides and free AuNPs. Bars represent the mean ± sd (n = 3). * *p* < 0.05, ** *p* < 0.01, *** *p* < 0.001, and **** *p* < 0.0001 denote statistically significant differences with respect to control (black column; basal activity of THP-1 macrophages without AuNPs).

**Figure 9 vaccines-13-00119-f009:**
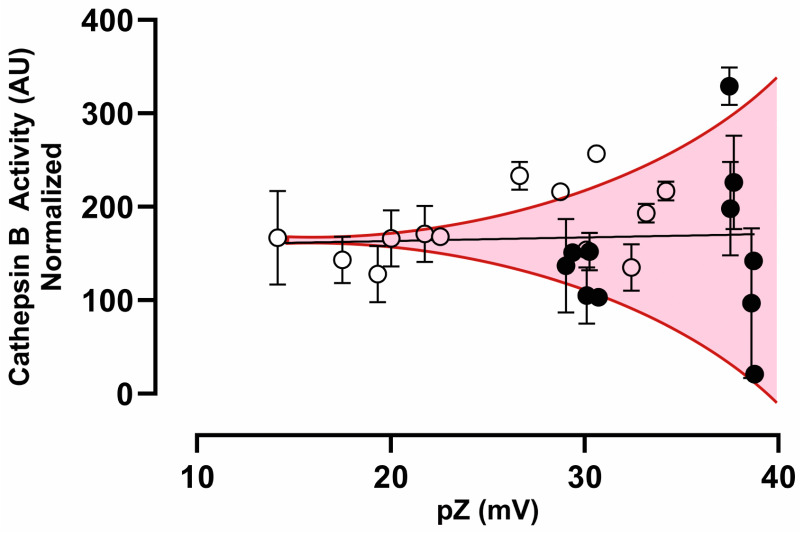
Correlation analysis between normalized cathepsin B activity and pZ values considering all AuNPs, including p-Au23 (open circle) and p-Au68 (filled circle). The normalization corresponds to the rate between cathepsin B activity and the percentage of cell survival (in arbitrary units). Each point corresponds to mean ± SD.

**Table 1 vaccines-13-00119-t001:** Peptide sequences conjugated to AuNPs, including a brief description of the inner regions.

Peptide	Sequence
P1	CALNNKKPKYVKQNTLKLAT
P2	CALNNKKPKYVKQNTLKLATRKKRQRRR
P3	CALNNGPGPGPKYVKQNTLKLAT
P4	CALNNGPGPGPKYVKQNTLKLATRKKRQRRR
Element	Description
CALNN	Nanoparticle anchoring region
PKYVKQNTLKLAT	Universal CD4+ T-cell epitope from haemagglutinin of influenza A virus
KK	Proteolytic motif for cathepsin B
GPGPG	Epitope spacer that favors immune processing
RKKRQRRR	Cell-penetrating peptide derived from the transactivator of transcription (TAT) of HIV

**Table 2 vaccines-13-00119-t002:** Putative sequences resulting from cathepsin B cleavage on peptides p1, p2, p3, and p4. In bold, the binding core of the CD4+ T-cell epitope predicted by Predivac-3.0 from the influenza A hemagglutinin is shown.

Peptide 1 Cleavage	Peptide 2 Cleavage	Peptide 3 Cleavage	Peptide 4 Cleavage
CALNNKKPKYVKQNT	YVKQNTLKLATRKKRQRRR	CALNNGPGPGPKYVKQNT	GPGPGPKYVKQNTLKLATRKKRQRRR
KKPKYVKQNTLKLAT	CALNNKKPKYVKQNTLK	GPGPGPKYVKQNTLKLAT	CALNNGPGPGPKYVKQNTLK
YVKQNTLKLAT	CALNNKKPKYVKQNT	PGPKYVKQNTLKLAT	CALNNGPGPGPKYVKQNT
CALNNKKPK	LKLATRKKRQRRR	CALNNGPG	LKLATRKKRQRRR
CALNN	LATRKKRQRRR	CALNN	LATRKKRQRRR
LKLAT	CALNNKKPK	LKLAT	CALNN

## Data Availability

The data presented in this study are available on request from the corresponding author.

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
