# Peer review of "Gold Nanoparticles as a Platform for Delivery of Immunogenic Peptides to THP-1 Derived Macrophages: Insights into Nanotoxicity"

_vaccines, 2025, doi:10.3390/vaccines13020119_

Round 1
Reviewer 1 Report
Comments and Suggestions for Authors
This manuscript describes the toxic effects of Au-NPs coated with immunogenic peptides to macrophages. The manuscript was well written, and experiments were well organized. The major issue with it is that the observed in vitro nanotoxicity might not reflect what really happens in vivo. This observed toxicity might be not even relevant to the vaccine efficacy/toxicity after injection. I would strongly suggest the authors address this in their manuscript before publication.
The use of NPs in vaccine delivery offers several advantages as described in the paper. However, the design principles for NP subunit vaccines are still far from clear. In this study, there are several designs that need to be clarified:
1. It is not clear why the authors choose macrophages for their studies. Depending on how these NPs are injected and trafficked in vivo, macrophage might or might not be the dominant antigen presenting cells.
2. The rationale of the design of peptide sequences needs to be addressed. For example, why cell-penetrating peptides sequences are needed and why cathepsin B targeting motifs are needed. In literature, it is still not clear whether an uptake or cleavage is actually needed in artificial antigen presentation. If the authors believe or have data to show that an effective uptake and antigen processing are needed for their system, they need to cite their papers (or data). Also, the requirements might be different between CD8 epitopes and CD4 epitopes.
3. Line 342 (Figure 1C and 2F) should be (Figure 1C and 1F).
Author Response
Comment 1: The major issue with it is that the observed in vitro nanotoxicity might not reflect what really happens in vivo. This observed toxicity might be not even relevant to the vaccine efficacy/toxicity after injection. I would strongly suggest the authors address this in their manuscript before publication.
Response 1: We thank the reviewer for this valuable observation. We agree that in vitro nanotoxicity findings may not match in vivo outcomes and do requires further verification in animal models to account for the physiological environment and variable complexities that may significantly alter the behavior of AuNPs, such as serum protein interactions, immune modulation and biodistribution. However, in vitro studies provide the background for understanding nanoconjugate uptake and toxicity mechanisms. Indeed, it is well-established that AuNPs accumulate in macrophages due to their strong phagocytic capacity, making these cells suitable to investigate the toxic effects of AuNPs. However, the functional impact of AuNPs exposure on immune system cells remains poorly documented, as the metabolism of macrophages and dendritic cells could be severely affected upon uptake of large amounts of these NPs and potentially leading to the alteration of cellular functions. Since our study aims to understand the design criteria influencing AuNP-based delivery of immunogenic peptides to THP-1 macrophages, we recognized the importance of investigating these interactions at the cellular level. We chose a well-established model for in vitro evaluation of immune responses [1], which enabled for precise control over key nanoconjugate design parameters and thus to improve our understanding of these variables and their effects over the cellular function. Such a level of control is difficult to achieve in vivo, where factors like biodistribution, immune system variability, and tissue-specific responses introduce more complexity. Therefore, in our understanding, initial in vitro results are useful and relevant to determine cellular-level effects of nanoconjugates and to guide future in vivo studies addressing the safety and effectiveness of AuNPs as immunogenic peptide delivery systems.
[1] Chanput, W.; Mes, J.J.; Wichers, H.J. THP-1 Cell Line: An in Vitro Cell Model for Immune Modulation Approach. International Immunopharmacology 2014, 23, 37–45, doi:10.1016/j.intimp.2014.08.002.
Accordingly, we have incorporated a new paragraph in the Discussion to address the relevance of in-vitro studies and highlighting the need to advance in-vivo studies (line 614 – 630):
“It is well-established that AuNPs accumulate in macrophages due to their strong phagocytic capacity, making these cells suitable to investigate the toxic effects of p-AuNPs nanoconjugates [57,58]. However, the functional impact of AuNPs exposure on immune system cells remains poorly documented, even though the uptake of large quantities of these NPs may severely alter cellular metabolism [59,60]. THP-1 macrophages are a well-described model for in vitro evaluation of immune responses [33,61], allowing to improve the understanding of key nanoconjugate design parameters and their effects on cellular function. Achieving such a level of control is difficult in vivo, since factors such as biodistribution, immune system variability and tissue-specific responses introduce additional complexity. While in vitro nanotoxicity findings may not always reproduce in vivo outcomes, these results shed light on the mechanisms involved in nanoconjugate uptake and toxicity at the cellular level. Furthermore, they are useful to guide future in vivo studies focused on the safety and effectiveness of AuNPs as a nanocarrier and delivery system for immunogenic peptides [62]. However, further research in animal models is required in order to consider the physiological environment and variables that could significantly alter the behavior of p-AuNPs, including serum protein interactions, immune modulation and biodistribution [63].”
Comment 2: The use of NPs in vaccine delivery offers several advantages as described in the paper. However, the design principles for NP subunit vaccines are still far from clear. In this study, there are several designs that need to be clarified: It is not clear why the authors choose macrophages for their studies. Depending on how these NPs are injected and trafficked in vivo, macrophage might or might not be the dominant antigen presenting cells.
Response 2: Thank you for highlighting this point. We selected THP 1-derived macrophages for the study because it is a well-established model for in vitro evaluation of immune responses. This cell line can be effectively differentiated into macrophages (professional APCs), and we confirmed the expression of macrophage-specific cell markers (Figure S2). We agree with the reviewer that macrophages may not be the primary antigen-presenting cells (APCs) in NP-induced immune responses. However, AuNPs accumulate both in macrophages and dendritic cells due to their phagocytic capacity, making these cells suitable to investigate the toxic effects of AuNPs. In addition, the role of lymph node-residing macrophages has been well-documented in mouse models injected with AuNP-based vaccines [2]. For instance, the immunological response of THP-1-derived macrophages was recently studied following treatment with gold nanorods [3], while AuNPs synthesized with both commercial and locally sourced honey were shown to downregulate IL-6 secretion in LPS-induced THP-1 macrophages [4].
[2] Kang, S.; Ahn, S.; Lee, J.; Kim, J.Y.; Choi, M.; Gujrati, V.; Kim, H.; Kim, J.; Shin, E.-C.; Jon, S. Effects of Gold Nanoparticle-Based Vaccine Size on Lymph Node Delivery and Cytotoxic T-Lymphocyte Responses. J Control Release 2017, 256, 56–67, doi:10.1016/j.jconrel.2017.04.024.
[3] Abuarqoub, D.; Mahmoud, N.N.; Zaza, R.; Abu-Dahab, R.; Khalil, E.A.; Sabbah, D.A. The In Vitro Immunomodulatory Effects of Gold Nanocomplex on THP-1-Derived Macrophages. Journal of Immunology Research 2022, 2022, 6031776, doi:10.1155/2022/6031776.
[4] Duncan, J.B.W.; Basu, S.; Vivekanand, P. Honey Gold Nanoparticles Attenuate the Secretion of IL-6 by LPS-Activated Macrophages. PLoS One 2023, 18, e0291076, doi:10.1371/journal.pone.0291076.
Therefore, we have edited and improved the paragraph at the end of the Introduction to clarify this point (lines 101-108):
“The goal of this study was to explore in a systematic manner the role of relevant design variables of p-AuNP conjugates on the viability of THP-1 macrophages, cellular uptake and endosomal cathepsin B activity in macrophages differentiated from the human monocytic leukemia cell line THP-1 (THP-1-derived macrophages), since this is a well-established cell model for in vitro evaluation of immune responses [33]. Therefore, AuNP concentration, AuNP size, peptide/CPP design and the effect of the peptide density on the NP surface were investigated, providing insight into the design criteria and nanotoxicology effects associated with AuNPs and immunogenic peptides in this specific cell line (Figure 1)"
Comment 3: The rationale of the design of peptide sequences needs to be addressed. For example, why cell-penetrating peptides sequences are needed and why cathepsin B targeting motifs are needed. In literature, it is still not clear whether an uptake or cleavage is actually needed in artificial antigen presentation. If the authors believe or have data to show that an effective uptake and antigen processing are needed for their system, they need to cite their papers (or data). Also, the requirements might be different between CD8 epitopes and CD4 epitopes.
Response 3: We thank the reviewer for the comment. From our perspective there is wide literature supporting the uptake and potential immunogenicity benefits of incorporating CPP sequences linked with both peptide antigens and NP-based peptide vaccines (nanoconjugates). However, despite the general assumption that CPPs have low toxicity, this is a serious concern (toxicity) regarding CPP applications in vaccine constructs and further studies are required to improve their delivery to specific cells types with reduced side-effects and enhanced efficacy (please see below a recent review). Given this, we sought to deepen our understanding of the specific cellular effects in THP-1-derived macrophages derived from incorporating the TAT sequence (a model CPP) at the C-terminus of the immunogenic peptides. Finally, we agree that peptides containing CD8+ or CD4+ T-cell epitopes should vary in terms of the motif/spacer sequences accounting for specific processing requirements (e.g. CD8+ T-cell epitopes are often designed with proteasome cleavage sites).
We have added a sentence in the Introduction to improve this point and also cited a recent review (32) in the manuscript (lines 97-100):
“In addition, the extent of induced toxicity and internalization mechanisms remain a matter of debate and further studies are required to improve their delivery to specific cell types with reduced side-effects and enhanced efficacy [30–32].”
[32] Hasannejad-Asl, B.; Pooresmaeil, F.; Takamoli, S.; Dabiri, M.; Bolhassani, A. Cell Penetrating Peptide: A Potent Delivery System in Vaccine Development. Front Pharmacol 2022, 13, 1072685, doi:10.3389/fphar.2022.1072685.
Regarding the observation on cathepsin B motifs, we agree with the reviewer this is a controversial subject, but we base our perspective on a rich body of literature supporting the use of cleavage motifs/linkers to improve the intracellular processing and efficacy of multi-epitope peptide-based vaccine constructs (please see examples: [5,6]).
[5] Behmard, E.; Soleymani, B.; Najafi, A.; Barzegari, E. Immunoinformatic Design of a COVID-19 Subunit Vaccine Using Entire Structural Immunogenic Epitopes of SARS-CoV-2. Sci Rep 2020, 10, 20864, doi:10.1038/s41598-020-77547-4.
[6] Jiang, F.; Han, Y.; Liu, Y.; Xue, Y.; Cheng, P.; Xiao, L.; Gong, W. A Comprehensive Approach to Developing a Multi-Epitope Vaccine against Mycobacterium Tuberculosis: From in Silico Design to in Vitro Immunization Evaluation. Front Immunol 2023, 14, 1280299, doi:10.3389/fimmu.2023.1280299.
Part of this background was previously included in the Introduction (lines 91-94): “CPPs have been fused in tandem with T-cell epitopes to successfully deliver immunogenic peptides into antigen presenting cells (APCs) [25,26]. In addition, this is a promising tool to improve intracellular delivery of nanomaterials [27,28]”.
However, we opted to improve a paragraph in the Discussion to provide additional context (lines 687-690):
“The bioinformatics design of the peptide sequence with cathepsin B-cleavable motifs was successfully achieved, based on the underlying assumption that this element can facilitate the correct release of the T-cell epitope for further processing through the MHC class II antigen presentation pathway [70, 71].“
Comment 4: Line 342 (Figure 1C and 2F) should be (Figure 1C and 1F).
Response 4: Thanks for pointing this out this mistake, which was now corrected in the manuscript.
Reviewer 2 Report
Comments and Suggestions for Authors
The article presents the development of gold nanoparticles for delivery of immunogenic peptides to macrophages. The topic of article is new and promising - many researchers are studying different nanoparticles as gene and drug carriers. The article is excellently written and easy to read and understand. The authors compared two kind of NPs of different size and made conclusions from this comparison. As results both nanoparticles were able to delivery peptides into cells. The data are correcly done and well described. The benefit of article is correct statistical nalysis. The data are very well discussed using good references. As for English, I am not native speaker, for me English is acceptable. I think the article can be published after minor corrections.
Minor:
Fig. 3 D. Please, do not use smooth lines.
Fig. 7D. Au-p68 , point 1.0 - Why it is so high? Please, explain.
Fig. 8 Please add error bars to points for better understanding of mechanisms.
Words in Title lines 8-12 "Affiliation 1 etc " can be removed.
Appendixes A and B - line 736 and below - can be removed.
Author Response
Comment 1: Fig. 3 D. Please, do not use smooth lines.
Response 1: Thank you for pointing this out. We deleted the smooth lines and have updated the legend of Figure 4D (previous Figure 3D) as it follow:
“(D) IC50 determination for Au23 (black) and p1-Au23 nanoconjugate (grey).” (lines 444-445)
Comment 2: Fig. 7D. Au-p68 , point 1.0 - Why it is so high? Please, explain.
Response 2: In our opinion it is challenging to provide a simple explanation to the increased activity observed in Figure 7B (p4-Au68 1.00), which we believe may result from a complex dynamic interplay between factors such as the resulting redox environment and pH changes generated by the internalized nanoconjugate. Previous works have shown that high levels of AuNPs can inhibit cathepsin B activity due to increased intracellular levels of ROS [1], which is the main mechanism driving gold nanoparticle toxicity and its oxidative effect on cysteine catalytic residues. However, the correlation between ROS and cathepsins extends beyond a simple cause-and-effect relationship in regulated cell death pathways such as apoptosis and autophagy [2]. While ROS promote apoptosis by inducing the release of cathepsins into the cytosol through disruption of lysosomal integrity, recent studies have suggested this process may also influence the subcellular localization of cathepsins [3]. For instance, cathepsins can translocate into the nucleus, where their activity appears to be enhanced during apoptosis [4,5]. In another study, Lagadic-Gossmann et al. observed changes in the expression and activity of cathepsins B and D in the rat pheochromocytoma cell line PC12 upon treatment with H2O2. Since cathepsins are known to activate at acidic pH levels within lysosomes, the authors of this study hypothesized that these proteases might also become active in the acidified cytoplasm during H2O2 and nitric oxide-induced apoptosis [6]. However, further investigations are necessary to unveil the mechanisms underlying these phenomena.
[1] Lalmanach, G.; Saidi, A.; Bigot, P.; Chazeirat, T.; Lecaille, F.; Wartenberg, M. Regulation of the Proteolytic Activity of Cysteine Cathepsins by Oxidants. Int J Mol Sci 2020, 21, 1944, doi:10.3390/ijms21061944.
[2] Voronina, M.V.; Frolova, A.S.; Kolesova, E.P.; Kuldyushev, N.A.; Parodi, A.; Zamyatnin, A.A. The Intricate Balance between Life and Death: ROS, Cathepsins, and Their Interplay in Cell Death and Autophagy. Int J Mol Sci 2024, 25, 4087, doi:10.3390/ijms25074087.
[3] Reiners Jr, J.J.; Caruso, J.A.; Mathieu, P.; Chelladurai, B.; Yin, X.-M.; Kessel, D. Release of Cytochrome c and Activation of Pro-Caspase-9 Following Lysosomal Photodamage Involves Bid Cleavage. Cell Death Differ 2002, 9, 934–944, doi:10.1038/sj.cdd.4401048.
[4] Meng, J.; Liu, Y.; Xie, Z.; Qing, H.; Lei, P.; Ni, J. Nucleus Distribution of Cathepsin B in Senescent Microglia Promotes Brain Aging through Degradation of Sirtuins. Neurobiology of Aging 2020, 96, 255–266, doi:10.1016/j.neurobiolaging.2020.09.001.
[5] Frolova, A.S.; Tikhomirova, N.K.; Kireev, I.I.; Zernii, E.Yu.; Parodi, A.; Ivanov, K.I.; Zamyatnin, A.A. Expression, Intracellular Localization, and Maturation of Cysteine Cathepsins in Renal Embryonic and Cancer Cell Lines. Biochemistry Moscow 2023, 88, 1034–1044, doi:10.1134/S0006297923070143.
[6] Lagadic-Gossmann, D.; Huc, L.; Lecureur, V. Alterations of Intracellular pH Homeostasis in Apoptosis: Origins and Roles. Cell Death Differ 2004, 11, 953–961, doi:10.1038/sj.cdd.4401466
Accordingly, we have incorporated a new paragraph addressing this explanation (lines 558 to 575):
“The increased cathepsin B activity observed in Figure 8D (p4-Au68 1.00) may result from a complex dynamic interplay between factors such as the resulting redox environment and pH changes generated by the internalized nanoconjugate. Previous works have shown that high levels of AuNPs can inhibit cathepsin B activity due to increased intracellular levels of ROS [50], which is the main mechanism driving gold nanoparticle toxicity and its oxidative effect on cysteine catalytic residues [51]. However, the correlation between ROS and cathepsins extends beyond a simple cause-and-effect relationship in regulated cell death pathways such as apoptosis and autophagy [52]. While ROS promote apoptosis by inducing the release of cathepsins into the cytosol through disruption of lysosomal integrity, recent studies have suggested this process may also influence the subcellular localization of cathepsins [53]. For instance, cathepsins can translocate into the nucleus, where their activity appears to be enhanced during apoptosis [54,55]. In another study, Lagadic-Gossmann et al. observed changes in the expression and activity of cathepsins B and D in the rat pheochromocytoma cell line PC12 upon treatment with H2O2. Since cathepsins are known to activate at acidic pH levels within lysosomes, the authors of this study hypothesized that these proteases might also become active in the acidified cytoplasm during H2O2 and nitric oxide-induced apoptosis [56]. However, further investigations are necessary to unveil the mechanisms underlying these phenomena.”
Comment 2: Fig. 8 Please add error bars to points for better understanding of mechanisms.
Response 2: We thank the reviewer for this comment. We have added error bars to each point as suggested and updated the legend of Figure 9 (previous Figure 8) as follow: “Each point corresponds to mean ± S.D.” (line 608).
Comment 3: Words in Title lines 8-12 "Affiliation 1 etc " can be removed.
Response 3: Thanks for pointing out this mistake, which was corrected in the manuscript.
Comment 4: Appendixes A and B - line 736 and below - can be removed.
Response 4: Thanks for pointing out this mistake, which was corrected in the manuscript.
Reviewer 3 Report
Comments and Suggestions for Authors
Eduardo Zúñiga et al. reported an interesting article about vaccine delivery through gold nanoparticles (AuNP). The topic fell within the scope of Vaccines. The study was well designed, and the manuscript well written. Overall, the manuscript deserved publication after a Minor Revision. Please refer to the following comments:
1. A Scheme depicting the whole picture of the study could be added at the end of the Introduction Section.
2. Section 2.12. Gold Enhance: Was this subtitle correct?
3. The PDI value of the nanoconjugates should be measured and reported.
4. Regarding Figure 8, the correlation function and its p value must be stated.
5. The statement of Supplementary Materials seemed to be incorrect.
Author Response
Comment 1: A Scheme depicting the whole picture of the study could be added at the end of the Introduction Section.
Response 1: We appreciate the reviewer´s comment, as the manuscript indeed lacked a clear description of the experimental strategy. Accordingly, we have incorporated a scheme that hopefully has clarified it and provides a better context to understand the research.
Comment 2: Section 2.12. Gold Enhance: Was this subtitle correct?
Response 2: We thank the reviewer for pointing this out. The subtitle was now corrected (line 260)
Comment 3: The PDI value of the nanoconjugates should be measured and reported.
Response 3: We thank the reviewer for raising this issue, as we realized we made a mistake in this point. The technique we have used to determine the size of the NPs was transmission electron microscopy (TEM) instead of dynamic light scattering (DLS), given that TEM is the gold standard for NP sizing and provides the diameter on the AuNP core. We have updated the subtitle to “Zeta Potential” (line 197). We apologise for this mistake.
Comment 4: Regarding Figure 8, the correlation function and its p value must be stated.
Response 4: Thank you for pointing this out. We have added the requested information in the text (lines 578-579).
Comment 5: The statement of Supplementary Materials seemed to be incorrect.
Response 5: Thanks for this valuable observation. The statement was corrected with the corresponding information from the Sup. material.